# How to Build and to Protect the Neuromuscular Junction: The Role of the Glial Cell Line-Derived Neurotrophic Factor

**DOI:** 10.3390/ijms22010136

**Published:** 2020-12-24

**Authors:** Serena Stanga, Marina Boido, Pascal Kienlen-Campard

**Affiliations:** 1Department of Neuroscience Rita Levi Montalcini, University of Turin, 10126 Turin, Italy; marina.boido@unito.it; 2Laboratory of Brain Development and Disease, Neuroscience Institute Cavalieri Ottolenghi, University of Turin, 10043 Orbassano, Italy; 3National Institute of Neuroscience (INN), 10125 Turin, Italy; 4Institute of Neuroscience (IoNS), Université Catholique de Louvain (UCLouvain), 1200 Bruxelles, Belgium; pascal.kienlen-campard@uclouvain.be

**Keywords:** neuromuscular junctions, neurotrophic factors, GDNF, neurodegenerative diseases, motor neuron diseases, AD, ALS, SMA, 3D neuromuscular model

## Abstract

The neuromuscular junction (NMJ) is at the crossroad between the nervous system (NS) and the muscle. Following neurotransmitter release from the motor neurons (MNs), muscle contraction occurs and movement is generated. Besides eliciting muscle contraction, the NMJ represents a site of chemical bidirectional interplay between nerve and muscle with the active participation of Schwann cells. Indeed, signals originating from the muscle play an important role in synapse formation, stabilization, maintenance and function, both in development and adulthood. We focus here on the contribution of the Glial cell line-Derived Neurotrophic Factor (GDNF) to these processes and to its potential role in the protection of the NMJ during neurodegeneration. Historically related to the maintenance and survival of dopaminergic neurons of the *substantia nigra*, GDNF also plays a fundamental role in the peripheral NS (PNS). At this level, it promotes muscle trophism and it participates to the functionality of synapses. Moreover, compared to the other neurotrophic factors, GDNF shows unique peculiarities, which make its contribution essential in neurodegenerative disorders. While describing the known structural and functional changes occurring at the NMJ during neurodegeneration, we highlight the role of GDNF in the NMJ–muscle cross-talk and we review its therapeutic potential in counteracting the degenerative process occurring in the PNS in progressive and severe diseases such as Alzheimer’s disease (AD), Amyotrophic Lateral Sclerosis (ALS) and Spinal Muscular Atrophy (SMA). We also describe functional 3D neuromuscular co-culture systems that have been recently developed as a model for studying both NMJ formation in vitro and its involvement in neuromuscular disorders.

## 1. NMJ Structural Architecture and Peculiarities

The development of the neuromuscular junctions (NMJs) is a complex chain of events in which many factors are intertwined in an extremely coordinated way. Briefly, the formation of NMJ requires first the guidance of the motor neuron (MN) axons toward the specific muscles for innervation and, secondly, the stabilization of the contact in order to form a mature NMJ. Indeed, this process of mature synapse formation is a key element for both the proper development and activity of the entire nervous system (NS). The neurotransmitter at the NMJ in skeletal muscle of vertebrates is Acetylcholine (Ach). Ach binds to nicotinic receptors (AchRs) on skeletal muscle fibers, thus inducing Na+ entry at the post-synaptic membrane, Ca++ release by the endoplasmic reticulum and finally fiber contraction. Non-myelinating terminal Schwann cells present at the NMJ contribute to its stabilization by capping the nerve terminal. Among macromolecules which participate in the synapse formation, the agrin-dependent pathway favors the clustering of AChRs. Indeed, initially dispersed along the muscle membrane, AChRs then slide towards the future NMJ and also local addition of AchRs are reported among the first events in NMJ formation. It involves the neuron-secreted proteoglycan agrin with its receptor, the lipoprotein receptor-related protein 4 (LRP4), the muscle-specific kinase (MuSK) and the regulatory soluble synapse-specific protease Neurotrypsin. Since MuSK is not able to directly bind Agrin, LRP4 mediates the process by forming a stable receptor/co-receptor assembly (for a complete review see [1]). The complex interplay between motor axons, terminal Schwann cells and the muscle fibers of the peripheral nervous system (PNS), is reminiscent of the tripartite synapse in the central nervous system (CNS) where pre- and post-synaptic neurons and astrocytes interact (Figure 1). Since defects in signal transmission between terminal nerve endings and muscle membranes are a common feature of several pathologic conditions, the neuro-muscular interplay represents a valuable model to study how the presynaptic message and the postsynaptic feedback collaborate to build a functional synapse, and how a pathological context can alter it.

## 2. Why Is It Relevant to Study the Neuro-Muscular Interplay in Neurological Disorders?

Neurological disorders have been historically approached as “central diseases” because of their pathogenesis specifically linked to the CNS. However, it is becoming increasingly evident and generally accepted that they should be studied from a holistic perspective [2,3]. The nervous system (NS) and its effectors such as muscles, which meet at the NMJs, should be considered as a whole when investigating pathogenetic mechanisms. We must also underline the contribution of the PNS to the maintenance and survival of neuronal tissue of the CNS through mechanisms of retrograde signaling. Indeed, growth factors secreted from muscles are retrogradely transported to the MN cell body thanks to the NMJ–muscle interplay. 

The correct formation and maintenance of the NMJ is fundamental for muscle function, i.e., also for vital and complex processes, such as breathing, feeding and locomotion. Therefore, muscle and NMJ degeneration also acquires a fundamental relevance in determining the quality of life and the autonomy of patients. Moreover, unproper NMJ formation with abnormal developmental synapse stabilization or improper NMJ maintenance during the progression of neuromuscular disorders, can trigger patient’s death because of respiratory failure. Neurological disorders, related to the aging of the population, but also to underlying diseases, represent an important health issue. To better understand and prevent neurological disorders, the PNS, and particularly the NMJ, could represent an easy and accessible model to reveal what is occurring in the CNS. Strong experimental evidence throughout time and recently developed 3D neuromuscular models highlighted the great potential of NMJ as a reliable and non-invasive model to study mechanisms involved in synaptogenesis [4]. Indeed, different animal models have provided basic fundamental milestones for the understanding of the guidance of growth cones and motor axons toward their muscle targets (many examples come from studies in Drosophila, C. elegans or mouse). 

## 3. Overview of the Neurotrophic Factor Support in the NS

Neurotrophic factors (NFs) are molecules secreted in order to support neuronal cells from their birth to differentiation into mature neurons [5]. Indeed, NFs promote cells’ development, differentiation and survival and, in the mature NS, they are critical for more complex activities such as the formation of synaptic plasticity and long-term memory. They have been largely studied for their potential therapeutic role in counteracting neurodegeneration [6]. However, their clinical use is limited because of difficulties especially related to protein delivery and pharmacokinetics in the CNS [7]. Moreover, NFs support the maintenance of complex structures (such as the NMJ) and their functionality; indeed, they are mostly—but not only—released by neuronal cells. Being soluble and diffusible factors, NFs work as inter-cellular communicators, which can be easily secreted and uptaken by cells. Depending on the specific secreting cell types and on the structure of the NFs, they are organized in superfamilies and families, whose sub-members possess peculiar characteristics and functions, which are summarized in Table 1 and described below.

The nerve growth factor (NGF)-superfamily, originally called neurotrophins, includes the nerve growth factor (NGF), the brain-derived neurotrophic factor (BDNF), neurotrophin-3 and 4 (NT-3, NT-4). These NFs can promote neuron survival by interacting with the neurotrophic tyrosine receptor kinases (Trks/NTR), or apoptosis and cell death by interacting with p75NTR [8]. NGF is the first member of the family that has been identified for its capability to induce neurite outgrowth in explants from sympathetic and sensory ganglia [9]. During development, the NGF superfamily and their receptors are present in axons and dendrites of growing neurons, in the trigeminal ganglion, and in pre- and postsynaptic terminals of neurons through adulthood. The most studied in relation to neurodegenerative processes are NGF and BDNF. NGF is highly produced by neurons within the central cholinergic system; changes in its expression have indeed been associated with an aging-and-brain-damage-dependent decrease in cognitive function [10]. BDNF is more specifically expressed in the hippocampus, cerebral cortex and amygdaloid complex [11]. Changes in its levels and activity in the CNS have been described in different neurodegenerative disorders [12]. Besides the CNS, peripheral tissues produce low concentrations of neurotrophins to maintain appropriate levels of neuronal survival and innervation. Indeed, unlike the CNS, once the junction is formed, it does not require neurotrophins signaling for the modulation of the neuromuscular transmission [13].

The transforming growth factor (TGF)-β superfamily is divided into three families: the glial-cell-line-derived neurotrophic factor (GDNF) family, the TGFβ family, and the bone morphogenetic protein (BMP) family. The first two families play key functions during NS development, and also the TGFβ family in T cell differentiation during the immune response [14], while BMPs are mostly studied because of their role in bone formation and little is known about their actions in the NS. TGFβ1 is expressed principally by glial cells in the CNS and recruited after brain damage, while TGFβ2 and 3 are found in many neuronal populations and in both astrocytes and Schwann cells in low concentrations [15]. The GDNF family includes GDNF and three structurally related members called neurturin, persephin and artemin. GDNF is a potent NF importantly involved in the formation and maturation of the neuromuscular synapse during development and disease. It is abundantly produced in skeletal muscles and Schwann cells, and, compared to the other NFs, its effects at the level of the NMJ are mediated through multiple pre- and postsynaptic mechanisms resulting in quite profound effects that are described in details in the next paragraph.

The neurokine or neuropoietin superfamily includes the ciliary neurotrophic factor (CNTF), the leukemia inhibitory factor (LIF), interleukin (IL)-6 and 11, cardio- trophin-1 (CTF1), the oncostatin-M, and the granulocyte colony-stimulating factor. All family members are related to the corresponding cytokines with whom they share similar tertiary structures. The neurokine family signals via the leukemia-inhibitory factor receptor (LIFR) and gp130 and is involved in the regulation of the neurotransmitter phenotype, neuronal and glial differentiation and development, and the rescue of neurons from axotomy-induced cell death [16]. CNTF is synthesized by muscles but it is not involved in NMJ maintenance; it prevents atrophy only in pathological states, such as in in response to nerve lesion [17].

Non-neuronal growth factors mainly include the fibroblast growth factors (FGF1 and 2), the epidermal growth factor (EGF) and the insulin-like growth factors (IGFs) FGF1 and 2, EGF and IGF1. They are present in large concentrations in the NS, where the most studied one is IGF-1. Together with its receptors, IGF-1 is expressed both in neurons and glial cells in the *substantia nigra* [18]. It is predominantly formed in the liver after stimulation by circulating growth hormone (GH) and released by the pituitary gland; it is also synthesized in peripheral tissues (such as bone, cartilage and skeletal muscle) but in low concentrations.

In this scenario, we focus on GDNF since it is the most closely related to NMJ because of its high expression in muscles and Schwann cells from development to adulthood; furthermore, since its contribution also in pathological conditions, it has been largely studied and tested in clinical trials for many neurodegenerative disorders [19].

## 4. The Role of GDNF at the NMJ

GDNF together with neurturin, artemin and peserphin is a sub-member of the GDNF family which is structurally related to the TGF-superfamily. Neurturin, artemin and peserphin are, respectively, involved in the support and survival of nigrostriatal neurons [20], sensory and sympathetic peripheral neurons [21] and of motor and midbrain dopaminergic neurons [21]. GDNF has been originally isolated from the conditioned media of a rat glioma cell line. Its first property as a trophic factor has been observed on primary cultures of dopaminergic neurons. It has been therefore largely studied for Parkinson’s Disease (PD) and evaluated in clinical trials [22,23]. As with all the other family members, GDNF acts by interacting with specific receptors to trigger different signaling cascades. Canonically, it forms a complex with GDNF family receptor alpha-1 (GFRα1), its primary receptor, which could be either membrane bound or soluble [24,25]. The so-formed complex activates the co-receptor tyrosine kinase REarranged during Transfection (RET) present on neuronal cell bodies and terminals and activates downstream pathways related to MAP kinases and Akt [26]. With a lower affinity, the GDNF–GFRα1 complex can signal via the RET-independent mechanism with the neural cell adhesion molecule (NCAM); GDNF can also signal independently from GFRα1 by interacting with the heparin sulfate proteoglycan syndecan-3 [27,28]. Originally thought to be secreted by glial cells, GDNF is also produced by neuronal cells in basal condition, while activated astrocytes are producing and releasing it in case of need, i.e., injury, inflammation and neurodegeneration [29,30]. Indeed, GDNF (which is mostly expressed during development) [31,32,33,34] is present in a healthy adult brain in the healthy adult striatal medium spiny neurons [35] and in the hippocampus [36], and it acts mainly on RET-expressing neurons. In MNs, in addition to the support of proliferation and maturation, GDNF participates in the regeneration of damaged axons and modulates the NMJ [37,38]. 

Moreover, GDNF is also expressed in cells located outside the CNS such as in peripheral nerve axons, Schwann cells [39] and skeletal muscles [40,41,42]. From there, it can be retrogradely transported to the soma of MNs as demonstrated by an in vitro model of the compartmentalized microfluidic neuromuscular co-culture system [43]. It also participates in NMJ formation and maintenance and its ability to induce neurite sprouting in axonal injury models has been clearly demonstrated [44,45]. Indeed, after spinal cord injury (SCI), GDNF is upregulated specifically in Schwann cells for weeks, making them important as potential points of intervention for SCI therapies [46,47]. Therefore, new attention and interest, relative to the neuronal population and to the mechanisms involved in GDNF secretion, is rising (for a recent review see [48]). 

GDNF can modulate the NMJ formation and functioning both at pre- and postsynaptic levels. Indeed, once secreted by muscles, GDNF is retrogradely transported to RET-expressing MNs [49], where it regulates presynaptic maturation. GDNF is one of the most potent survival factors for MNs [37,50]. In vitro experiments demonstrated its ability to increase the amplitude and the frequency of spontaneous synaptic currents [51], while in mice overexpressing GDNF or after GDNF administration the NMJs are hyperinnervated [52]. Moreover, in muscles/MNs co-cultures, GDNF and Neurturin are able to (i) enhance the aggregation of ACh synaptic vesicles at the presynaptic terminal areas facing the NMJ, (ii) enhance neurotransmitter release and iii) induce AChR clustering at the postsynaptic terminal [53]. A more recent study demonstrated with an elegant microfluidic chamber model that GDNF promotes axon growth, only when it is applied to the NMJ, and not to the soma [43]. Indeed, GDNF can influence the maturation of the postsynaptic site and can also induce myocyte differentiation. In muscle cell lines, GDNF expression favors myoblasts differentiation into myotubes and spontaneous contraction in vitro [54]. In basal conditions, GDNF participates to the maintenance of the junction and to its continuous remodeling by stimulating the endplate size and complexity [53,55,56]. GDNF is also an activity-dependent factor and its secretion is elicited in case of damage, in order to promote pro-survival mechanisms, or as a positive outcome of physical exercise involving muscles. Indeed, GDNF expression relies on the recruitment of the myofibers during physical activity [57,58]. Moreover, when GDNF is overexpressed in muscle, an increase in the number of fast-twitch and fatigue-resistant (type IIa) fibers is observed, and this phenomenon correlates with functional tests [34]. On the contrary, it has been observed that muscles that are not actively recruited during exercise display no changes in NMJ morphology, and GDNF levels remained similar to those in sedentary controls [59]. 

To summarize, GDNF has been shown to be (i) the most abundant factor released by both skeletal muscles and Schwann cells; (ii) to be secreted at the NMJ during development and also adulthood; (iii) to be a fundamental NF for the formation of NMJ both at the pre-synaptic site—for ACh vesicles aggregation—and at the post-synaptic site—for AChR clustering; (iv) to be a neuromodulator able to exert long-term regulatory effects on synaptic transmission; (v) to exert a protective role during peripheral degeneration occurring in neurodegenerative disorders.

## 5. The NMJ-GDNF Cross-Talk: What Goes Wrong During Neurodegeneration? 

In different progressive and severe neurodegenerative processes, besides the CNS, there is also a clear involvement of peripheral tissues, especially of mitochondria-enriched muscles and NMJs. Their structural impairment and loss of function, often occurring before symptom onset, increase progressively during degeneration. These early phenomena are largely described both in animal models and in patients (for a recent review see [60]). Because of the significant involvement of the GDNF at the NMJ, we highlight here the GDNF/NMJ cross-talk in the context of severe disorders such as Alzheimer’s disease (AD), Amyotrophic Lateral Sclerosis (ALS) and Spinal Muscular Atrophy (SMA). 

AD, the most common form and cause of dementia, is a major social health problem and, the in absence of a meaningful treatment, it remains a major scientific and medical challenge. The neurodegenerative process has always been studied with a ‘central’ approach following its pathological hallmarks—amyloid plaques and tau neurofibrillary tangles—which accumulate in vulnerable CNS regions. Related to amyloid deposits, it has been recently observed that GDNF is able to decrease proinflammatory mediators exerting an anti-inflammatory function in AD. Indeed, GDNF treatment is able to control amyloid-beta (Aβ)-induced inflammatory response in microglia [61]. Moreover, the role of GDNF in astrocytes, oligodendrocytes and microglia in regulating tissue clearance from Aβ deposition in the brain is increasingly studied [62]. Interestingly, AD-related proteins such as the Amyloid Precursor Protein (APP) family members—APP and the two paralogs Amyloid Precursor-like Protein1 and 2 (APLP1 and APLP2)—have been associated to NMJ formation. Indeed, APP family members work as transsynaptic regulators: MN APP interacts with Lrp4 expressed at the level of the muscles, promoting MuSK activation and NMJ formation and stabilization [63,64]. Muscular APP controls the expression of the GDNF involved in the maintenance of the NMJ [54]. Presenilins (PSs) are the catalytic subunit of the gamma-secretase involved in Aβ release upon APP processing [65,66,67]. More recently, a discrimination between PS1 and PS2 catalytic properties has been established [68], and their involvement in cellular bioenergetics and mitochondrial respiration [69] supports their potential role in muscle trophism and NMJ formation. 

ALS and SMA are mainly categorized as MN disorders. Besides MN degeneration, they are largely characterized by muscle and NMJ degeneration. Although both are considered rare disorders, on one side it has been estimated that by 2040 around 400,000 patients will be diagnosed with ALS worldwide [70], and on the other SMA it is the leading cause of infant mortality due to genetic causes (i.e., mutation of the survival motor neuron-1—SMN1—gene) [71]. 

In ALS, GDNF is playing an important protective role at the level of the NMJ. Indeed, bilateral administration of GDNF into the tibialis anterior, the forelimb triceps brachii and into the long muscles of the dorsal trunk in a familiar rat model of ALS, preserves the NMJ, promotes MN function and survival [72], and muscle-specific GDNF overexpression increases the lifespan of an ALS mouse model [73]. On the contrary, systemic injection of AAV9-GDNF to young rats, in a model of ALS, did not increase lifespan; even though AAV9-GDNF injection resulted in modest functional improvement, important sides effects related to systemic injection (i.e., slower weight gain, reduced activity levels and decreased working memory) were observed [74]. Interestingly, in ALS, MN degeneration is probably preceded by NMJ denervation. Indeed, in muscles from both mouse models of familial ALS and in ALS patients, APP is upregulated and its expression correlates with the progression of clinical symptoms [75,76]. Moreover, GDNF and soluble APP fragment levels are altered at the onset of motor deficits in the SOD1 transgenic mutant mouse model and in the cerebrospinal fluid (CSF) of patients [77]. Overall, this piece of evidence suggests a link between the ongoing denervation process and the attempt to counteract it by both GDNF and NMJ-related interactors, such as APP. 

Concerning SMA, despite its well-known genetic origin, until 2017 no treatment was available. Nusinersen (Spinraza; Biogen), an antisense oligonucleotide, is the first drug approved by the Food and Drug Administration (FDA) (in December 2016) and by the European Medicines Agency (EMA) (in June 2017) for both infants and adults with SMA. Recently, two other drugs have been approved: Zolgensma (AVXS-101), an adenovirus by Avexis/Novartis (May 2019—FDA; May 2020—EMA), and Evrysdi (Risdiplam), a new oral drug by Genentech, a member of the Roche Group (August 2020—FDA). These drugs aim at restoring SMN levels [78]; however, since also in SMA NMJ degeneration is an early event, addressing the NMJ could represent a potential and adjunctive therapeutic target (for a review, see [79]). Indeed, NMJ degeneration in SMA occurs in parallel to the degeneration of mitochondria in spinal MNs (Figure 2). These organelles have been described as altered and fragmented, with compromised functional activity also in humans [80], impacting in turn on muscle trophism. Many groups described immature, small and fragmented NMJs in the most used SMA animal model, the SMNΔ7 mouse, compared to controls [81,82,83]. These defects precede the onset of motor symptoms underlying the importance of NMJ degeneration in the pathology [84,85]. Furthermore, one of the hallmarks of SMA pathology is the accumulation of neurofilament (NF), the cytoskeletal components of neurons, at the NMJ, highlighting its central role in the pathology. Indeed, NF accumulation at the NMJ results in altered organelles and trophic factor dynamics, negatively impacting on disease progression [82,86]. Few reports describe the alteration of GDNF levels in different tissues/fluids in SMA models. GDNF levels have been described as reduced in SMA-induced pluripotent stem cell (iPSC)-derived astrocytes, but lentivirus expression of GDNF in the model is not able to recover astrocytic processes or calcium signaling response [87]. In the CSF of SMA patients, GDNF levels, but not NGF and BDNF levels, were importantly increased compared to controls and authors suggested that it could be the results of a GDNF attempt to counteract the loss and damage of neuronal cells [88]. Considering the impairment of the NMJ in SMA and the role of GDNF at that level, further studies about its role in tissues more relevant to the pathology (such as the spinal cord, muscles and the NM-interplay) should be undertaken. Models of nerve–muscle co-cultures, which we describe in the paragraph below, could indeed help in the finding of new potential pathways involving the NMJ in SMA.

## 6. 3D Neuromuscular Models for NMJ Reconstruction 

Recently, experimental approaches based on 3D models to rebuild the neuromuscular contact in vitro have been developed and optimized. These models are useful not only to study the properties of maturing muscle fibers, but especially to unveil adult human NMJ formation and development. Indeed, NMJ maturation is a process requiring both muscle fibers and motor neurons; therefore, 3D approaches where muscle cells are co-cultured with MNs are needed for better investigating disease mechanisms and applying them to neurodegenerative models. 

Different models can be set up by using (i) cell lines of muscle cells (C2C12 cells) and cholinergic or MN models (usually NG108-15, NSC-34), (ii) primary cells obtained from animal models (myoblasts and primary MNs or embryonic-derived spinal cord slices) or 3D co-cultures composed of (iii) human induced pluripotent stem cells (iPSCs) of myoblasts and MNs. Each model possesses its own characteristics and can be chosen and adapted in order to answer to the specific scientific question and/or applied to the peculiar disorder. 

(i)3D models performed with cell lines have the great advantage to be both handy and straightforward. Because of the dividing rate of the cells, NM contacts are rapidly obtained after 7/8 days of culture. This model is particularly suitable for studies requiring silencing and/or overexpression of specific genes in order to mimic a pathological condition. Indeed, muscle and neuronal cell lines can be efficiently transfected in co-culture models [54]. However, it is important to note that manipulation of the cells via transfection could affect itself the formation of the NM contact and, furthermore, that the transfection process is only transient.(ii)When NM contacts are performed with primary myoblasts and MNs or spinal cord slices, the timing for obtaining NM contacts is longer and can reach up to a month of culture [89]. Furthermore, a very precise coordination for animal crossings is needed in order to have pups at postnatal day P3-P5 for myoblasts cultures and, around 12–14 days after culture, embryos at E12,5 for MNs or spinal cord slices. On the other side, by working with cells/tissues derived from the animal model of interest, the disease is recapitulated without the need for further manipulation.(iii)Recently, studies based on the co-culture of human iPSCs have been implemented. The major advantage of these techniques is to perform cultures with patient-derived cells (usually fibroblasts) that are specifically reprogrammed in order to generate iPSCs to be directly differentiated into muscle or MN cells (for a comprehensive review on the topic, see [90]). Two recent works have been published regarding 3D neuromuscular models based on iPSCs. The first model, by Bakooshli and colleagues consists in mixing human muscle progenitors with human pluripotent stem cell-derived MNs [91]. The co-cultures are able to self-organize to form functional NMJ connections. Authors validated the functional connectivity by calcium imaging and electrophysiological tests, and they applied the model to the study of myasthenia gravis. The second model, by Osaki and colleagues, is based on muscle and MN interaction through a microfluid device [92]. This model needs weeks to be implemented, but the advantage is that it permits to record in real time the NMJ formation and the synchronization of MN activity and muscle contraction.

It would be interesting to deepen the role of GDNF in 3D models for NMJ formation in health and disease. Such techniques represent the fundament for developing precision medicine and drug screening directed to the NMJ for neuromuscular disorders. 

## 7. Conclusions

NMJ formation and maintenance require neurotrophic support. In this fine-tuned machinery, GDNF plays a pivotal role both in NMJ assembly and in maintaining the signaling between MN and muscles and vice versa. Indeed, GDNF supports a tightly coordinated assembly of pre- and postsynaptic structures and a strong reciprocal signaling between the two counterparts. The role of GDNF turns out to be decisive from development to aging passing through adulthood by promoting neuroplasticity and preventing from neurodegenerative disorders. Looking at neurodegenerative disorders as multisystemic disorders, the involvement of the periphery in the study of pathogenic mechanisms could represent a more successful approach also for finding new targets for therapies. Especially in the case of dementia, where a precise diagnosis is difficult in the early disease phase, researchers are largely studying peripheral biomarkers obtained from easily accessible peripheral fluids—such as blood—and by non-invasive methods [93,94,95]. Interestingly, since GDNF expression induced in response to exercise participates to the rejuvenation of the NMJ structure and function [59], it could be evaluated as a possible peripheral marker and possible targets. Furthermore, it could play a key role in delaying the onset of aging and neurodegenerative disorders by preventing NMJ degeneration. The study of the GDNF-NMJ cross-talk in recently developed 3D models and its monitoring could indeed be translated into accurate exercise and a healthy lifestyle prescription to elderly human subjects and patients with the aim to preserve the integrity of the neuromuscular system through aging and disease [96].

## Figures and Tables

**Figure 1 ijms-22-00136-f001:**
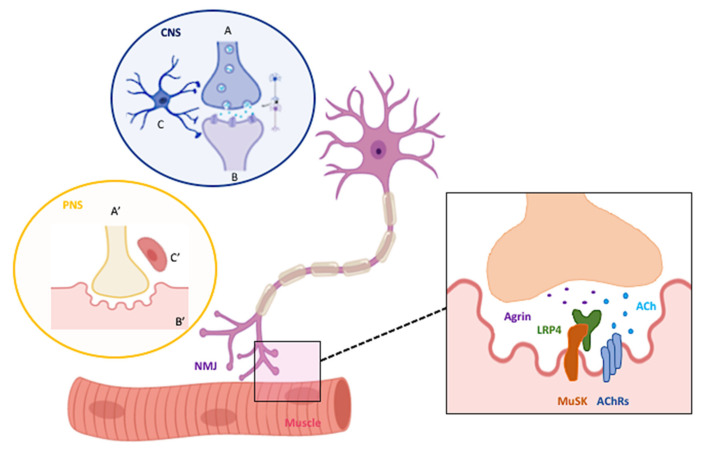
The central vs. the peripheral tripartite synapse. The NMJ is at the crossroad between the NS and the muscle, representing their cross-talk point. Central and peripheral synapses share a similar morphological feature: the tripartite synapse. Indeed, in the CNS the tripartite synapse is composed by a presynaptic axon (**A**), a postsynaptic dendrite (**B**) and by astrocytes (**C**). In the PNS, there is a comparable structure and the tripartite communication is between the axon terminal of the MN (**A′**), the muscle fiber (**B′**) and the terminal Schwann cells (**C′**). The macromolecules involved at the nerve–muscle interplay are represented in the zoomed section. Created with BioRender software.

**Figure 2 ijms-22-00136-f002:**
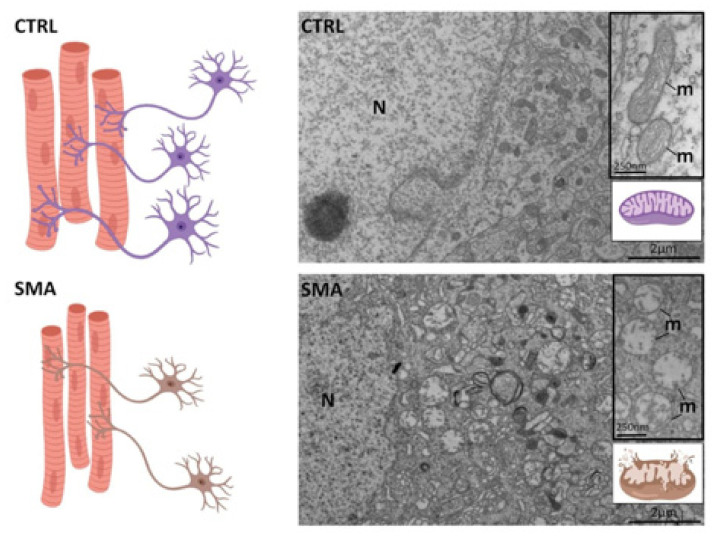
Peripheral dysfunctions in SMA. SMA is characterized by progressive MN death, impaired innervation and muscular atrophy, contributing to NMJ destabilization. Among the known pathogenic mechanisms, mitochondrial alterations represent an early event. As shown in the Electron Microscopy images from spinal cord samples of the SMNΔ7 mice (postnatal day 10), compared to the control (CTRL), SMA mitochondria are clearly altered, dilated and fragmented. N = nucleus; m = mitochondria. Created with BioRender software.

**Table 1 ijms-22-00136-t001:** NF superfamilies, families and sub-members’ major functions and expression at the peripheral level. Human NFs are listed in the table. Symbols + or + + are used to indicate the degree of expression and level of secretion in peripheral tissues under physiological conditions: + low levels; + + high levels. The expression levels in peripheral tissues of the NFs in bold (GDNF, CNTF and IGF1) are the most representative of the family.

Superfamilies	NFs Families	Submembers	Families’ Major Functions	Expression in Peripheral Tissues
NGF	NGF BDNF NTF3 NTF4		NS development, Neuron survival or apoptosis and cell death, Neurite outgrowth	Skeletal muscle +
TGF-β		**GDNF**		
		Neurturin		
		Artemin	NS development,	Skeletal muscle ++
	GDNF family	Persephin	NMJ formation and maintenance	Schwann cells ++
	TGFβ family			
	BMP family	TGFβ1-3	NS development	Immune system ++
				Schwann cells +
		BMP 1-6, 71, 8a, 8b, 10, 15	Bone formation	Bone ++
Neurokine or Neuropoietin	**CNTF** LIF IL-6 IL-11 CTF1 Oncostatin-M Granulocyte colony-stimulating factor Cardiotrophin-1		NS development, Regulation of neurotransmitter phenotype, Neurons’ rescue from axotomy-induced cell death	Skeletal muscle +
Non-Neuronal Growth Factors	FGF1 FGF2 EGF **IGF1**		NS survival	Liver ++ Bone + Cartilage + Skeletal muscle +

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
