# Peer review of "How to Build and to Protect the Neuromuscular Junction: The Role of the Glial Cell Line-Derived Neurotrophic Factor"

_ijms, 2020, doi:10.3390/ijms22010136_

Round 1

Reviewer 1 Report

This review is overall well-written with some issues denoted below.

Major comments

The abstract suggests that the literature and support for GDNF’s role in treating diseases is much more clear than it appears to be. However, in the manuscript the authors seem to be making a case for GDNF as a therapeutic, in repairing diseased and dead neurons to treat degenerative neuronal diseases such as Alzheimer’s disease.   Thus, the abstract should be adjusted to reflect this emphasis.

Minor comments

p. 3, line 104: “uptaken up” is redundant

p. 3, line 116 and line 117: “GFRα1” and “RET” are not defined

p. 3, line 119: insert “a” before RET-independent

p. 3, line 122: “and from while 122 activated. . .” does not make sense

p. 4, line 128: “. . . pre- and postsynaptically levels” does not make sense

p. 4, line 138: “level” --> levels

p. 7, line 268: “. . . the genetic of the disease” does not make sense.

Author Response

This review is overall well-written with some issues denoted below.

Dear Reviewer,

thank you very much for the time you dedicated to revise our manuscript and for the positive report.

We carefully considered both the major and the minor points that you highlighted. As you can see in the new version we submitted with the track-change mode, we corrected the manuscript accordingly. 

Major comments

The abstract suggests that the literature and support for GDNF’s role in treating diseases is much more clear than it appears to be. However, in the manuscript the authors seem to be making a case for GDNF as a therapeutic, in repairing diseased and dead neurons to treat degenerative neuronal diseases such as Alzheimer’s disease.   Thus, the abstract should be adjusted to reflect this emphasis.

Major comment: 

The abstract has been modified in order to better reflect the content of the manuscript (i.e. the therapeutic potential of GDNF for peripheral degenerative processes occurring in neurodegenerative disorders).  

Minor comments

p. 3, line 104: “uptaken up” is redundant

p. 3, line 116 and line 117: “GFRα1” and “RET” are not defined

p. 3, line 119: insert “a” before RET-independent

p. 3, line 122: “and from while 122 activated. . .” does not make sense

p. 4, line 128: “. . . pre- and postsynaptically levels” does not make sense

p. 4, line 138: “level” --> levels

p. 7, line 268: “. . . the genetic of the disease” does not make sense.

Minor comments:

We checked for English and we corrected all the misspellings you indicated through the entire manuscript. 

Reviewer 2 Report

This review article centers around the idea that GDNF critically mediates the formation, function and long-term maintenance of the NMJ, with GDNF dysfunction driving synaptic dysfunction and loss in disease.  However, it is not clear how the authors settled on GDNF, since evidence for this neurotrophic factor is no more compelling than others (e.g., BDNF, VEGF) that similarly promote development and function of the NMJ over the lifespan, and are also similarly implicated in neurodegenerative disease.   Moreover, the review article seems to lack substance, with the discussion also sometimes poorly focused.  One notable example of poor focus is the discussion of the gut-brain axis and its likely role in disease.  Its relevance to GDNF and/or NMJ is not clear.  While the title indicates that there will be "lessons" learned from GDNF about its role in NMJs, it is difficult to discern what these promised lessons are.  An interesting exercise for the authors of this manuscript might be to list these putative lessons; such lessons could then provide a structure for the manuscript.  In sum, it seems that a much more in-depth analysis of the literature on neurotrophic factors and their role in NMJs is needed, and would be a valuable contribution to the field. In its present form, however, its value is not clear.  GDNF is only one of many likely candidates that regulate NMJ formation and function, with likely have a role in disease.

Author Response

Dear Reviewer,

thank you very much for the time you dedicated to revise our manuscript and for the constructive report. We carefully considered your observations, and we agree with your comments. As you suggested and as you can see in the new version we submitted with the track-change mode, we extended on the role of neurotrophic factors in the NS. By making this, it emerged clearly why to focus on GDNF when speaking about NMJ. We believe that in this form the review is more comprehensive and that now it could be a valuable contribution to the field. More in details: 

  • We performed an in-depth analysis of the literature on the role of neurotrophic factors and we added a new paragraph (see section 3 – line 96) with an overview of their roles in the NS. We added a new Table (Table 1) where we highlight their localization and major function. We added 15 new references; 
  • Focusing on the processes of neurodegeneration occurring in the periphery, we centered on NMJs as a good model to study peripheral phenomena and on GDNF contribution at this level. The reasons why to focus on GDNF now emerge and are recapitulated at the end of paragraph 4 – line 182. Furthermore, we changed the title of the review in order to be softer and more aligned with the focus;
  • Regarding the therapeutic potential of GDNF, we agree with the Reviewer that it is not the only putative candidate. However, since the focus of the review is the role of GNDF at the NMJ, we specifically deepened on its potential for therapy against peripheral neurodegeneration. Interestingly, it emerges that the major actors of the CNS pathologies that we describe are either controlling GDNF expression in muscles (i.e. APP in AD knock-out mice models), or overexpressed in neuromuscular disease models at very early stages (i.e. APP again in ALS mice models, and prior to denervation in muscles from ALS patients). Moreover, GDNF substation therapies have been indeed foreseen in PD, even though it’s no more tested, as we mentioned in the text (exogenous GDNF) and recent papers on the topic have been cited. We also highlight the fact that endogenous GDNF levels has been suggested as a biomarker (for ALS and also SMA) and its measure could be an interesting read-out for evaluating the therapeutic efficacy especially for the pediatric disorder by non-invasive methods;
  • We checked for English through the entire manuscript and we reviewed the discussion by correcting/eliminating out-of-focus sections (such as the gut-brain axis as suggested by the Reviewer – line 78) which, indeed, could have been misleading for readers.